# Synthesis and Characterization of TiO_2_ Thick Films for Glucose Sensing

**DOI:** 10.3390/bios12110973

**Published:** 2022-11-05

**Authors:** G. Silva-Galindo, M. Zapata-Torres

**Affiliations:** Instituto Politécnico Nacional, CICATA Unidad Legaria, Legaria 694, Mexico City 11500, Mexico

**Keywords:** titanium dioxide (TiO_2_), biosensor, non-enzymatic, glucose, electrochemical

## Abstract

In this paper, we present the results of a non-enzymatic electrochemical glucose biosensor based on TiO_2_. An anatase working electrode was synthesized using the spin coating technique with the polymeric precursor method and dispersed TiO_2_ nanoparticles. Through scanning electron microscopy, it was observed that the electrode presented an irregular surface with clusters of nanoparticles. Electrochemical characterization indicated that the response was directly related to the morphology of the electrode. In the presence of glucose, the electrode exhibited adsorption behavior toward the molecules, enabling their recognition. The electrode was tested by employing PBS (phosphate buffer solutions) with varying pH values (from 4 to 9), demonstrating its electrochemical stability, even in the presence of glucose. Amperometric characterization was used to determine that the working region appeared from 0.2 mM to 2 mM, with a sensitivity of 4.46 μAcm^−2^mM^−1^ in PBS pH 7. The obtained results suggest that TiO_2_-based electrodes could be used for the detection of glucose concentration in sweat (0.277–1 mM) and saliva (0.23–1.77 mM).

## 1. Introduction

Diabetes mellitus is a chronic disease derived from an endocrine disorder. It appears when the pancreas does not produce enough insulin for the human body (type 1 diabetes) or when the insulin produced in the pancreas is not used effectively by cells (type 2 diabetes) [1]. In recent years, diabetes has been considered an epidemic [2]; its onset and progression may be slow in origin and asymptomatic, leading to secondary complications or emergent lethal symptoms. The early diagnosis and monitoring of the disease require the control of glucose levels in the body in order to delay and even prevent the progression of microvascular complications, such as retinopathy, nephropathy, neuropathy, and macrovascular complications, such as stroke and coronary heart disease [3,4].

Some hospitals and laboratories are capable of detecting glucose in clinical blood samples through the usage of spectroscopies [5,6,7]. Nevertheless, the demand for glucose monitoring systems for self-testing is constantly increasing, owing to the increment of people with diabetes. In 2019, diabetes affected 463 million people worldwide; the number of people with this disease is expected to increase to 700 million by 2045 [8]. The rapid growth of nanotechnology has allowed the biosensing industry and researchers to develop various glucose recognition processes and analyses for portable devices, among which electrochemical sensors stand out [9]. Their reliability is attributable to the transduction method, which is considered one of the most promising detection methods, owing to its sensitivity, response time, selectivity, portability, and ease of use [10].

Biosensors are composed of three miniaturized screen-printed electrodes: the reference, counter, and working electrode, which is enabled by a specific enzyme on its surface. The procedure to identify glucose levels uses the fingerstick method. First, a drop of blood is collected on the working electrode; then, the blood reaches the enzymes and is recognized by an oxidation-reduction process; finally, the produced current is processed through a sensing platform, which provides the glucose concentration in the sample [11]. Enzymes offer high selectivity during glucose measurement. Despite their dominance in the glucose biosensing industry, their use is inherently limited by their high cost of production and the enzyme denaturation, which generally results from use and environmental changes related to temperature, humidity, and pH [12]. These critical flaws make enzymatic biosensors single-use, disposable devices. To overcome these disadvantages, new generations of biosensors have been introduce [10].

Non-enzymatic biosensors avoid the use of enzymes; they promote the execution of glucose recognition processes directly on the surface of the working electrode. This next-generation technology has enabled the development of glucose biosensors with improved detection limits and glucose measurement in non-blood samples [13].

It is well-known that the electrode materials play a critical role in the construction of non-enzymatic electrochemical biosensors. The direct glucose recognition process varies depending on the electrode material used. In the last decade, extensive research efforts have been focused on the construction of functional electrode materials, enhancing catalytic activity, stability, sensitivity, conductivity, and biocompatibility to facilitate the signal transduction process and to amplify the electrochemical signal produced by the redox recognition event.

Noble metals, such as gold and platinum, have been extensively used in the construction of working electrodes [14]. These materials can oxidize glucose molecules through the chemisorption process [15]. The electrodes execute the glucose oxidation process in an alkaline solution environment at a pH 13. However, subjecting the electrodes to physiological pH conditions, in which glucose can be found from pH 4.5 to pH 8.1 [9], results in a poisoning effect by chloride and phosphate ions. When such an event occurs, the rate of glucose oxidation decreases proportionally to the ion concentration, resulting in a current response that overestimates the glucose concentration in the sample. Sodium chloride is an essential compound in physiological fluids. It is present in high concentrations in blood, with values ranging from 98 to 106 mM [15].

Some other metals, such as nickel, copper, zinc, and their oxide forms, have also been used in biosensor applications [15]. In these materials, the redox reaction of transition metal centers can explain the oxidation process of glucose [16]. Such materials are highly resistant to chloride ion poisoning. However, they show low electrical conductivity and a narrow linear detection range below the average glucose level in blood (2–40 mM) [9] and well above glucose levels in fluids, such as saliva (0.24–1.67 mM) [17], sweat (0.01–1.1 mM) [18], and tears (0.05–5 mM) [9]. Moreover, outside of alkaline solution environment, their functionality is lost.

In recent years, titanium dioxide (TiO_2_) has been the focus of a number of publications, owing to its extensive application potential. In the biomedical context, it is used for the construction of prostheses and bone coatings [19] because it is an economic material that can be synthesized using multiple techniques. Thus, various morphologies can be achieved, such as fibers, nanotubes, spherical particles, and thin and mesoporous films [20]. A simple technique to synthesize TiO_2_ thick films involves a spin-coating system with presynthesis of disperse nanoparticles using the polymeric precursor method [21]. The advantages of this technique include the preparation environment, which does not require an inert atmosphere. Furthermore, varying temperatures and proportions between citric acid and nanoparticles enable the control of stoichiometry, purity, agglomerates, and toxicity [22,23]. Moreover, the thickness of the films can be managed by adjusting the number of deposit layers.

The aim of the present investigation was to evaluate the reliability of a non-enzymatic electrochemical biosensor based on TiO_2_ to determine the glucose concentration in a sample using the spin coating technique. The novelty of this work is based on the study of a pure TiO_2_ working electrode in the anatase phase toward glucose recognition, in addition to the study of electrode stability in response to changes in pH buffer solution, which represents a barrier to the development of glucose biosensors, as well as the study of the sensibility of the electrode through double-pulse chronoamperometry to evaluate its application as a non-invasive glucose biosensor.

## 2. Materials and Methods

### 2.1. Synthesis of a TiO_2_ Electrode

First, a 2 × 2 cm^2^ Titanium (Ti) foil sample (purity: 99.9% Ti; 0.25 mm thick) from Sigma Aldrich Co. was used as a support. It was polished with 0.1, 0.3, and 0.05 μm α-alumina; before modification, it was subjected to sonication in distilled water, then subjected to a sonication in isopropanol, acetone, and hexane for 15 min each at an operating frequency of 40 kHz.

Secondly, two layers of TiO_2_ were deposited on the Ti substrate by the spin coating technique, using commercial P25 TiO_2_ anatase nanoparticles synthetized with a Pechini solution. The Pechini solution was prepared from a titanium isopropoxide/citric acid/ethylene glycol solution with a molar ratio of 1:4:16, respectively. The solution was prepared by adding ethylene glycol and titanium isopropoxide into a volumetric flask and heated to 85 °C with stirring for one hour. Then, citric acid was added, and the solution was stirred at this temperature until it turned clear. The solution was deposited on the Ti substrates employing the spin coating technique at 1500 rpm for 60 s. Finally, after depositing the TiO_2_ layers, the Ti/TiO_2_ electrode was subjected to heat treatment in a Thermo Lindberg blue HTF55667C quartz tube model at 450 °C for one hour under argon flow [24].

### 2.2. Characterization of the Ti/TiO_2_ Electrode

To identify the morphology of the synthesized Ti/TiO_2_ electrode, a Jeol JSM-6390LV scanning electron microscope (SEM) was employed with magnifications of ×1000 and × 5000. The Ti/TiO_2_ electrode was optically analyzed with diffuse reflectance spectroscopy. To calculate the value of its bandgap energy the Kubelka–Munk method was employed [25]. UV-vis characterization was carried out with an Ocean Optics USB4000 + XR1-ES spectrometer with a conversion range of 190–1050 nm and a DH-2000 integrated lamp.

The thickness of the electrode was measured at 10 points with a Mitutoyo 543-132 instrument, obtaining a value of 125 ± 12 µm. The crystalline structure of the electrode was analyzed by X-ray diffraction (XRD) technique using a Bruker D8 Advance diffractometer with a Bragg Brentano geometry, sweeping from 10° to 70° with a step of 0.05° per second under Cu-Kα radiation, with an acceleration voltage of 40 KeV and a current of 25 mA.

All electrochemical measurements were performed using a VersaSTAT 3/Autolab computerized potentiostat/galvanostat (Princeton Applied Research). A conventional system of three electrodes was used in all electrochemical measurements; a 1 cm^2^ platinum mesh was used as an auxiliary electrode, with a double-junction Ag/AgCl 3.8 M KCl reference electrode and the synthesized Ti/TiO_2_ electrode as the working electrode with a geometric contact area of 0.78 cm^2^. Solutions were prepared using a phosphate buffer solution (PBS) at varying pH values (pH 4, pH 7, and pH 10) and varying concentrations of D(+)-glucose. The buffer solutions and D(+)-glucose were purchased from Sigma Aldrich Co. Cyclic voltammetry measurements were carried out in a potential range of −1 V to 1 V, whereas the double-pulse amperometric curves were obtained at −1 V and −600 mV by adding 0.2 mM of glucose successively. All measurements were performed at room temperature; all chemicals were of analytical grade and used without further purification.

## 3. Results and Discussion

### 3.1. SEM

The images obtained through scanning electron microscopy are shown in Figure 1. Fissures on the surface of the electrode are clearly seen in Figure 1a, which are potentially attributable to the heat treatment to which the electrode was subjected during the synthesis procedure. Figure 1b shows the surface under increased magnification, with regions of agglomerated TiO_2_ particles.

To study the effect of cracks and agglomerated particles on the reproducibility of the electrodes, four two-layer working TiO_2_ electrodes were constructed and electrochemically tested; then, the error related to the synthesis procedure was calculated.

### 3.2. UV-Vis

The diffuse reflectance spectrum of the synthesized Ti/TiO_2_ electrode is shown in Figure 2a. The films presented a high absorption region for wavelength values of less than 400 nm, corresponding to the ultraviolet region.

Figure 2b presents the bandgap of the electrode calculated by the Kubelka–Munk method. The results show that the electrode has a bandgap of approximately 3.29 eV. The obtained bandgap value is close to that reported for TiO_2_ synthesized through dispersed nanoparticles [15]. Because the Ti/TiO_2_ electrode was synthesized from TiO_2_ nanoparticles in the anatase phase and further subjected to thermal treatment at 450 °C, the bandgap value is associated with the crystallinity of the film.

### 3.3. XRD

Figure 3 displays the X-ray diffraction pattern of the synthesized Ti/TiO_2_ electrode. The pattern was indexed using the JCPDF (Joint Committee powder diffraction file).

Most Ti/TiO_2_ diffraction peaks were matched to the anatase phase corresponding to the (101), (004), (200), (105), (211), (204), (116), (220), (215), and (303) planes according to card number 211,272. The XRD pattern showed only one phase in the sample, without impurities in the crystalline structure.

### 3.4. Electrochemical Measurements

#### 3.4.1. Cyclic Voltammetry (CV)

The voltammogram presented in Figure 4 shows the curves recorded when the electrode was immersed in a pH 7 PBS. The electrochemical response evidences three regions. Region (i) represents the behavior of the Ti/TiO_2_ electrode, similar to that of an ideally polarizable electrode, in which Faradaic processes do not occur. In this region, under the imposition of positive potentials, electrostatic attractions prevail at the electrode–solution interface. Regions (ii) and (iii) are recognized as electronically active and can be described as follows. In region (ii), when the cathodic potential sweep begins, it promotes an accumulation of negative charge on the electrode surface; such accumulation is compensated by cations in the PBS solution, such as Na^+^ and K^+^. When the cathodic potential sweep is reversed, region (iii) appears; in this region, because potentials move toward less negative values, the accumulated charge is returned, and the cations are desorbed. This process leads to a maximum of anodic current density with a positive value.

The process that occurs in region (ii) near to the exponential growth of charge accumulation was associated with the quasi-reversible filling of the monoelectronic states (traps) located below the conduction band [26]. In some semiconductor oxides, the presence of traps is related to structural defects produced by doping processes. However, in TiO_2_, it is attributed to the surface of the electrodes, especially to the presence of grain boundaries [27]. The filling processes of the energy traps were previously described by G. Boschoo and D. Fitzmaurice [28], who reported that the occupation of these energy states can be described by chemical Equations (1) and (2), occurring simultaneously with the charge compensation accumulated in the conduction band.
(1)$$Ti^{IV}{}_{}-OH+e^{-}+\left(M^{+}o\;H^{+}\right)\;\to \;Ti^{III}{}_{}-OH{\left(M^{+}o\;H^{+}\right)}_{}$$
(2)$$O^{-}{}_{}+\left(M^{+}o\;H^{+}\right)\;\to \;O_{}\left(M^{+}o\;H^{+}\right)$$

When the electrode was immersed in a pH 7 PBS and glucose, its electrochemical response changed. As shown in Figure 4, in region (ii), a decrease in the cathodic current density occurred. Furthermore, in region (iii), an increase in the anodic current density appeared.

In this work, we propose that the changes produced in the electrochemical response of the electrode may be attributable to electrostatic interaction between glucose molecules and the charge accumulated in the traps. In the synthetized Ti/TiO_2_ electrode, we suggest that traps are located throughout fissure regions and between the agglomerated particles observed in the SEM images. Such traps enable the adsorption of glucose molecules through electrostatic interactions.

Figure 5 shows a proposed mechanism of electrode–glucose interaction. When the cathodic swept potential begins, in region (i), the charge begins to accumulate in traps; then, as an effect of electrostatic interaction, glucose molecules move toward the electrode and are adsorbed. This adsorption allows for the rapid occupation of trap states; when glucose molecules are adsorbed, they can be potentially oxidized, enabling charge transfer with TiO_2_. As a consequence, the cathodic current density decreases.

However, not all electronic states of the traps are occupied by glucose molecules. Owing to their size, their mobility is reduced compared to that of the ions present in the solution. Such ions move faster toward the surface of the electrode, compensating for the accumulation of charge in the traps. During the anodic sweep potential, in region (ii), glucose products and ions are desorbed, leading to an increase in anodic current. Further characterization should be employed to assert charge transfer between TiO_2_ and glucose in real time [29].

##### Scan Rate Effects

Cyclic voltammetry measurements of the Ti/TiO_2_ electrode were performed in pH 7 PBS with 1 mM of glucose at varying scan rates from 10 to 110 mVs^−1^. Figure 6a shows the CV curves. The shape of the curves does not change evidently, even at higher scan rates. The current densities increase with increasing potential scan rates, indicating a satisfactory rate property.

The linear plot of anodic current (jpa) vs. the square root of velocity (υ) suggests that the electrode process is diffusion-controlled; the ions that compensate for the accumulated charge on the surface of the electrode flow to the electrode from a region of lower to higher concentration until homogeneity is achieved. The linear equation for anodic current can be expressed as follows: j_pa_ = 42.94υ + 9.24; R^2^ = 0.9813. Figure 6c shows the relationship between the logarithm of the anodic peak current vs. the logarithm of the scan rate; the linear equation is expressed as: log j_pa_ = 0.4708 log υ + 1.6931; R^2^ = 0.9792. The slope is relatively close to the theoretical value of 0.5 for a purely diffusion-controlled process. To maintain a reasonable electrode response, a scan rate of 10 mV/s was chosen for further characterization.

##### Influence of Solution pH

The electrochemical behavior of the Ti/TiO_2_ electrode was tested using cyclic voltammetry, with varying PBS pH values from 4 to 9 and a constant concentration of glucose of 1 mM. Figure 7a shows the obtained voltammograms. The cathodic and anodic current density decreases as the pH increases. Both currents decrease with decreased H^+^ ion concentration or increased OH^−^ ion concentration. The CV patterns also show the variation of the potentials related to the variations in pH solution. Figure 7b depicts the potential at which the cathodic current growth begins. The potential decreases with respect to the increase in pH. To study these effects, the following linear equation for cathodic current vs. pH was obtained: E_ca_ = −0.604 pH + 0.2696; R^2^ = 0.9855. The value of the slope is close to the theoretical value of 0.058 VpH^−1^ for a two-electron and two-proton process [30]. Such behavior has been reported as a universal characteristic of some semiconductor oxides in the presence of an aqueous solution [31].

When evaluating the change in potential with respect to the pH of the solution, we assumed that the obtained slope is related to the displacement of the TiO_2_ edge conduction band. Initially, when the electrode interacts with the solution, its energy at the Fermi level must be balanced with the energy of the solution’s redox potential so that equilibrium occurs through a process of charge carrier transfer between the semiconductor and the solution. Electrons flow takes place until a thermodynamic equilibrium in the interface is achieved. When an external negative potential is imposed on the electrode, an accumulation of electrons in the conduction band is promoted, which generates a region of charge enrichment in the semiconductor. This accumulation leads to a conduction band deflection toward more negative potentials and is compensated for ions in the solution. When the negative potential is reversed, the ions are desorbed, as shown in the following chemical equation:(3)$$Ti^{IV}O_{2}+e^{-}+H^{+}\;↔\;Ti^{III}\left(O\right)\left(OH\right)$$

Under this condition, the net charge on the TiO_2_ surface can be either positive, zero, or negative. This net charge is important because it influences the location of the energy edge of the conduction band [26]. As shown in Figure 7a, under acidic pH values, the *Ti^IV^* reduction process begins at more positive potentials so that a greater amount of oxide is reduced to *Ti^III^*. This process allows for increased conduction band charge accumulation, which is compensated for by the insertion of Na^+^, *K*^+^, and *H*^+^ ions and results in an increase not only in the cathodic current density but also the anodic current density. 

In basic pH conditions, the opposite occurs. The conduction band is shifted to more negative potentials, so the reduction process limits the accumulation of charge and, consequently, the growth of cathodic and anodic current density. Despite differences related to current density owing to the change in the pH solution, the Ti/TiO_2_ electrode was not corroded. We can assert the aforementioned because after each CV test, the electrode did not present with physical deterioration. Furthermore, during the CV measurements, it preserved its electrochemical response at pH 7, as well as at higher and lower pH values. This is an advantageous result, as some pH body fluids are found within this interval, such as sweat (4.5–7) and saliva (6.2–7.6) [9].

#### 3.4.2. Chronoamperometry

The double-pulse chronoamperometric measure is presented in Figure 8a, with a magnification of the second pulse shown Figure 8b. A proportional relationship is evident with the current density produced by the increase in glucose concentration in the concentration range of 0.2 mM to 2 mM.

The increase in cathodic current density is related to the accumulation of charge at the electrode–solution interface. Figure 9a shows an increase in accumulated charge on the electrode surface with respect to the increase in glucose concentration, which indicates that the glucose presence contributes to the accumulated charge compensation in the TiO_2_ conduction band. Figure 9b shows that a two-layer Ti/TiO_2_ electrode is able to detect glucose concentrations between 0 and 2 mM in a pH 7 PBS with a sensibility of 4.52 μAcm^−2^mM^−1^. Our proposed electrode is ideal for the detection of glucose in fluids, such as sweat and saliva, in which glucose concentrations range between 0.06 mM and 1 mM and 0.23 and 1.77 mM, respectively [9].

#### 3.4.3. Reproducibility and Artificial Saliva Study

Four working electrodes were constructed to study the effect of the synthesis process. Figure 10a represents the current cathodic density changes. The standard deviation of current in the four electrodes is 2.70 μAcm^−2^, with a standard error of 1.35 μAcm^−2^. The results of the present study confirm the reproducibility of the electrodes, even when employing a handmade procedure.

Furthermore, the Ti/TiO_2_ electrode was tested in artificial saliva prepared according to the procedure and reactants described in [32]. Figure 10b shows the increase in current density with increased glucose concentration. The changes related to glucose concentration vary in a narrow current interval, from −2 μA to −1 μA; however, there are notable differences between the current concentrations. Further research should be conducted enhance the sensibility of the electrodes, potentially including the study of TiO_2_ electrodes with varying morphologies.

## 4. Conclusions

In this work, we developed a non-enzymatic electrochemical glucose biosensor based on TiO_2_. Our investigation highlights that the glucose recognition process based on the use of TiO_2_ electrodes occurs mainly as a result of the formation of traps during the synthesis procedure, in combination with the pseudocapacitive properties of TiO_2_ and the propensity of glucose molecules to adsorb on the surface of the electrode. The pseudocapacitive properties of TiO_2_ and the adsorption of glucose molecules occur in parallel during the imposition of the semiconductor working potentials, enabling quantification of the presence of glucose. The reported results demonstrate that among the advantages of using TiO_2_, the regeneration capacity of the material stands out because it does not involve corrosion processes during glucose measurement, even with changes in the pH of the solution from 4 to 9. The proposed non-enzymatic glucose biosensor based on a Ti/TiO_2_ electrode is not only easy to synthesize with an economical technique, such as spin coating, but also able to determine low glucose concentrations in samples.

## Figures and Tables

**Figure 1 biosensors-12-00973-f001:**
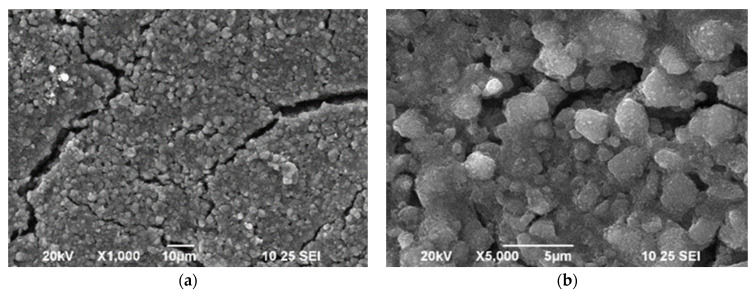
Morphology of the Ti/TiO_2_ electrode with under (**a**) ×1000 and (**b**) ×5000 magnification.

**Figure 2 biosensors-12-00973-f002:**
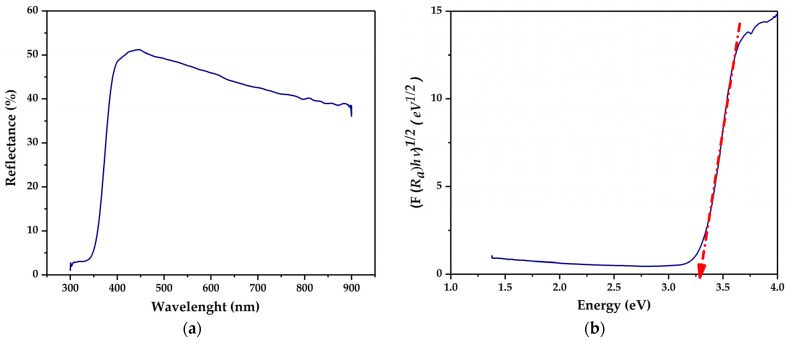
(**a**) Diffuse reflectance spectrum; (**b**) relation of (F (Rα) hν)^1/2^ vs. the energy of the Ti/TiO_2_ electrode.

**Figure 3 biosensors-12-00973-f003:**
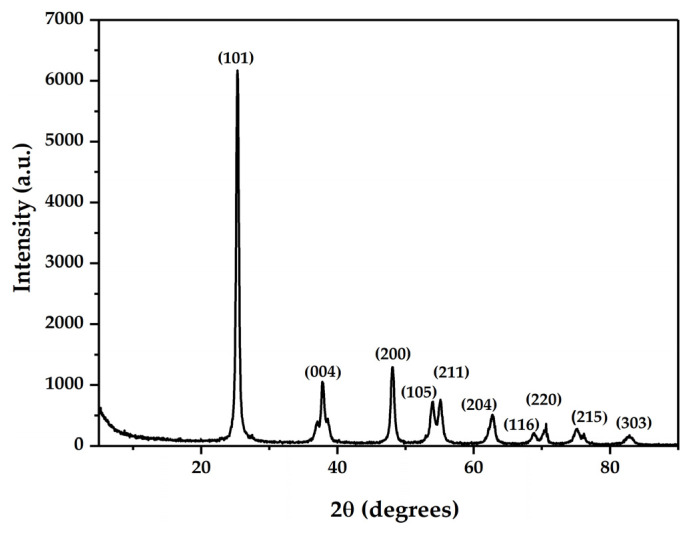
X-ray diffraction pattern of the Ti/TiO_2_ electrode synthesized by spin coating.

**Figure 4 biosensors-12-00973-f004:**
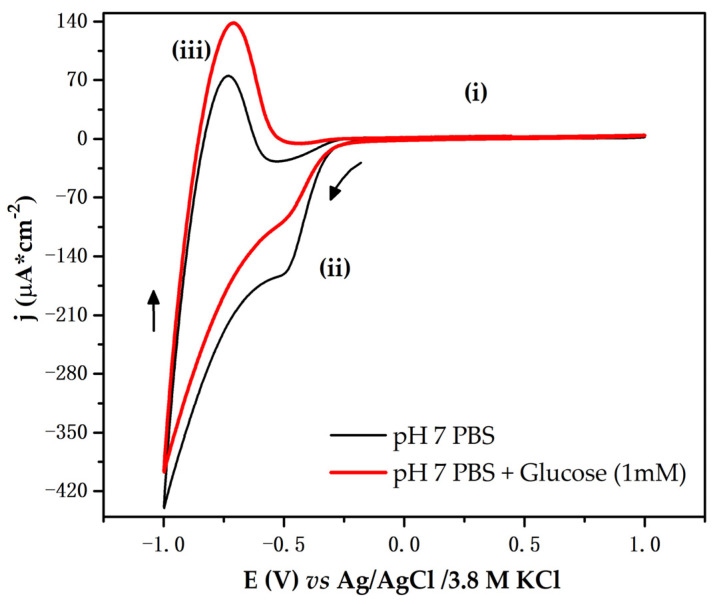
CV curves registered for the Ti/TiO_2_ electrode recorded at 10 mV/s in pH 7 PBS and pH 7 PBS with D(+)-glucose at 1 mM. The CV electrochemical response is characterized by region (i) where Faradaic processes do not occur. Regions (ii) and (iii) represent the reduction and oxidation process of TiO_2_, respectively. Arrows depict the sense of potential sweep.

**Figure 5 biosensors-12-00973-f005:**
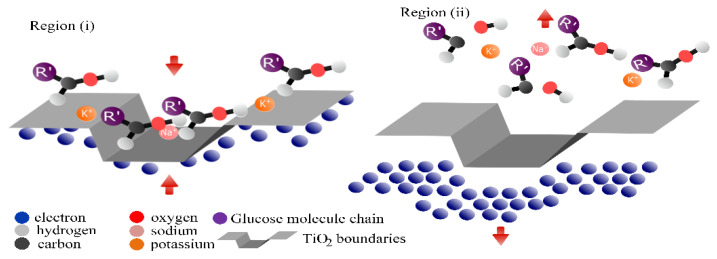
Proposed mechanism of interaction between ions, glucose molecules, and TiO_2_ grain boundaries throughout the CV process.

**Figure 6 biosensors-12-00973-f006:**
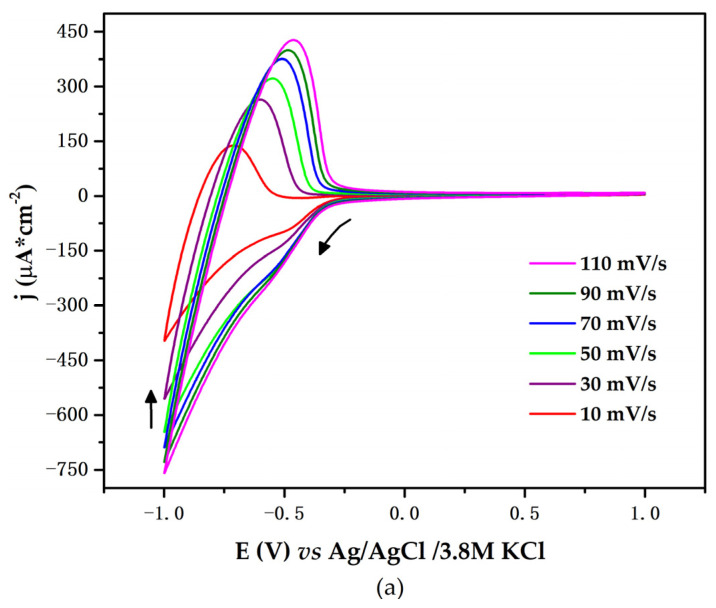
(**a**) CV curve of the working electrode under the effect of scan rate variation (10, 30, 50, 70, 90, and 110 mV/s), using pH 7 PBS and glucose at 1 mM. Arrows depict the sense of potential sweep. (**b**) Plot of anodic peak current versus square root of scan rate. (**c**) Plot of log anodic peak current versus log scan rate.

**Figure 7 biosensors-12-00973-f007:**
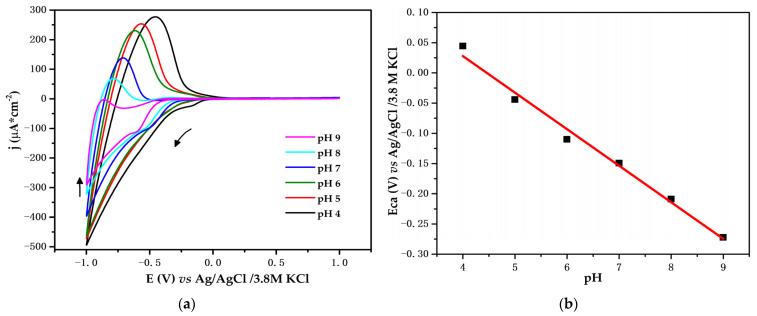
(**a**) Working electrode at varying pH values of PBS with glucose at 1 mM; *v* = 10 mV/s. Arrows depict the sense of potential sweep. (**b**) Relationship between cathodic potential vs. pH.

**Figure 8 biosensors-12-00973-f008:**
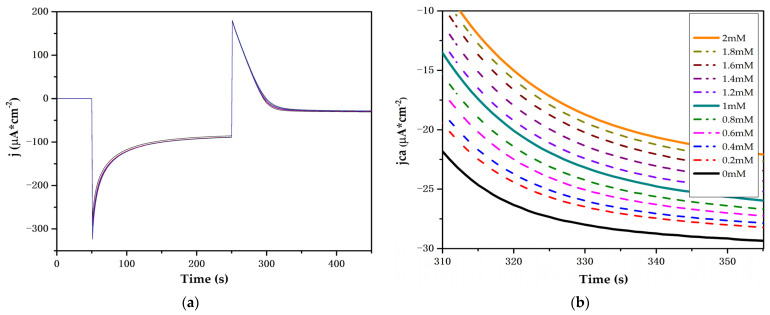
(**a**) Ti/TiO_2_ electrode response to double-pulse chronoamperometry in a pH 7 PBS and glucose from 0 to 2 mM. (**b**) Magnification of the second chronoamperometric pulse region.

**Figure 9 biosensors-12-00973-f009:**
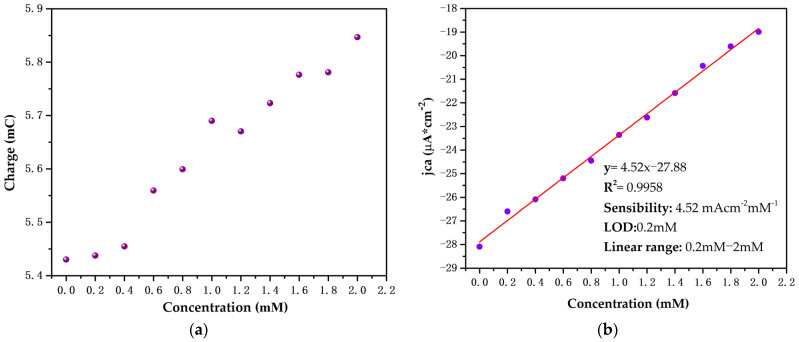
(**a**) Calculated charge accumulated on the electrode surface during reduction process. (**b**) Determination of sensitivity and limit of detection (LOD) of the synthesized electrode at 330 s.

**Figure 10 biosensors-12-00973-f010:**
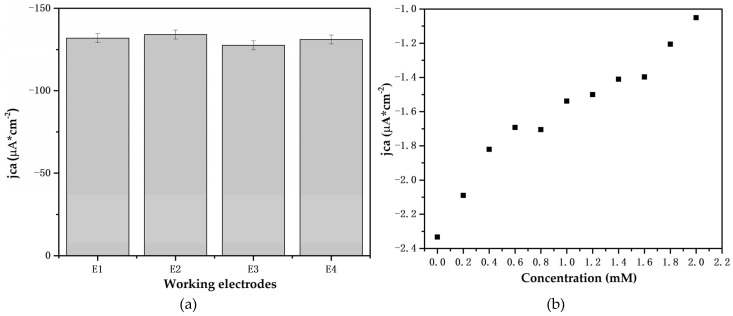
(**a**) Cathodic current density of the five Ti/TiO_2_ electrodes in pH 7 PBS and glucose at 1Mm. (**b**) Cathodic current density of a Ti/TiO_2_ electrode in artificial saliva solution at 330 s.

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
