# Peer review of "Synthesis and Characterization of TiO2 Thick Films for Glucose Sensing"

_biosensors, 2022, doi:10.3390/bios12110973_

Round 1
Reviewer 1 Report
Please see attached file.

Author Response
The reply is as a comments in the pdf file

Reviewer 2 Report
The manuscript by Silva-Galindo and Zapata-Torres reports the characterization of a TiO2 based non-enzymatic electrochemical sensor for glucose detection.The paper is well enough written, apart from minor details, and the experimental characterization sounds pretty solid.
I recommend publishing the work after the following have been addressed:
- Fig.4. Panel a is redundant.
- Line 170: typo
- Please, rephrase sentence 178-182.
- Please, rephrase sentence 187-190 and comment it.
- The assumption made in line 192 197 should be discussed in detail and supported by reference.
- Line 221-221: these deductions should be discussed more in detail. And checked for
- Fig7b. I’m a bit confused regarding y axis. Is it the open circuit potential? Some comments regarding ph7, which is the working condition of the sensor should be made.
Author Response
We correct what was suggested by the reviewer in the text of the article.

Reviewer 3 Report
In the submitted manuscript, the authors developed a biosensor based on TiO2 nanoparticles for glucose concentration measuring in the body’s fluids, especially in sweat and salvia.
Contrary to the other biosensors based on TiO2 nanoparticles whose response is based on a direct electrochemical signal, the biosensor constructed in this manuscript, for the first time, used an indirect electrochemical response. The response is based on the pseudo-capacitive properties of TiO2 and the propensity of glucose molecules to adsorb to the surface of the electrode. This is the main novelty of this work. The authors completely characterized obtained electrode and found that defects in the electrode’s surfaces influence the response of the biosensor. The authors investigated the biosensor’s ability for use in a wide range of pH and the presence of different glucose concentrations. They found that constructed biosensor could be used in sweat and salvia.
I would like to stress that the manuscript is interesting and should be published in this journal. However, before final acceptance, the authors should test their biosensor in real samples and point out its advantages and disadvantages.
In addition, I found that the manuscript’s title is very similar to the one published last year in the same journal (https://doi.org/10.3390/bios11050149, Non-Enzymatic Glucose Biosensor Based on Highly Pure TiO2 Nanoparticles).
I have suggested that the authors change it because the readers could think that the same work is presented. I have to stress that although the title of these manuscripts are similar and both constructed biosensors detect glucose based on TiO2, the working principle differ between themselves.
Author Response

(The authors gave the same response as above.)

Round 2
Reviewer 1 Report
All the comments of reviewer were properly addressed. The manuscript can be accepted for publication.